# Paraneoplastic Amyotrophic Lateral Sclerosis: Case Series and Literature Review

**DOI:** 10.3390/brainsci12081053

**Published:** 2022-08-08

**Authors:** Zhao Yang, Lu He, Manli Ren, Yizhou Lu, Huanyu Meng, Dou Yin, Sheng Chen, Qinming Zhou

**Affiliations:** 1Department of Neurology, Ruijin Hospital, Shanghai Jiaotong University School of Medicine, Shanghai 200240, China; 2Institute of Neurology, Shanghai Jiaotong University, Shanghai 200240, China; 3Ruijin Hospital Neurology Training Base, Shanghai 200240, China; 4Co-Innovation Center of Neuroregeneration, Nantong University, Nantong 226019, China

**Keywords:** paraneoplastic amyotrophic lateral sclerosis, SOX1, GAD65, case series, literature review

## Abstract

Paraneoplastic amyotrophic lateral sclerosis (ALS) is a rare and special type of ALS. The pathogenesis, clinical presentation, treatment and prognosis remain poorly understood. We herein presented three cases of paraneoplastic ALS. In case 1, we first reported an ALS patient with the positive serum antibodies against both Sry-like high mobility group box 1 (SOX1) and glutamic acid decarboxylase 65 (GAD65). However, immunotherapy did not improve his neurological symptoms. We also reported two ALS patients with renal clear cell carcinoma and chronic myelogenous leukemia. No positive paraneoplastic antibodies were detected in either the serum or the cerebrospinal fluid of the two patients, and their clinical symptoms progressed slowly after tumor treatment. The three cases enriched the existing case pool of this rare disorder. In addition, we have comprehensively reviewed the literature of paraneoplastic ALS. The clinical features, treatment effect and prognosis were summarized to broaden our understanding of paraneoplastic ALS.

## 1. Introduction

Amyotrophic lateral sclerosis (ALS) is a fatal and rare neurodegenerative disorder characterized by degeneration of the upper and lower motor neurons of the brain and spinal cord [1,2,3]. Although ALS is not a classical phenotype of paraneoplastic neurological syndromes (PNS), PNS are one of a multitude of causes of ALS [4]. Although they are rare, the clinical situations that support the diagnosis of definite paraneoplastic ALS are: laboratory evidence of well-characterized onconeural antibodies or neurological improvement is observed after cancer treatment [4,5,6]. When onconeural antibodies are absent, if the onset is rapidly progressive or there are inflammatory findings in the CSF or brain/spine MRI, and cancer present within two years of diagnosis or neurological symptoms remained stable after tumor treatment, it supports the diagnosis of possible paraneoplastic ALS [4,5,6]. Paraneoplastic ALS generally has low incidence rates; hence, there are currently few case reports on the condition, and the description of the clinical characteristics of paraneoplastic ALS patients remains incomplete.

Therefore, this article aimed to raise the awareness on paraneoplastic ALS by reporting three cases and conducting a literature review.

## 2. Case Series

Case 1: A 50-year-old male was admitted to the hospital with 10 months of progressive weakness on the right upper limb, accompanied by slurred speech. He noticed atrophy of his right hand muscles and developed dysarthria seven months after onset. Neurological examination revealed marked fasciculation in his atrophic tongue and diminished bilateral gag reflexes. Muscle strength testing showed moderate weakness in the right limbs. Moreover, the deep tendon reflexes on the right upper limb were brisk, and the bilateral Babinski sign was positive. Sensory disturbance was absent. Brain and spine magnetic resonance imaging (MRI) were unremarkable. Electromyography (EMG) indicated low compound motor action potentials (CMAPs) in the bilateral median, ulnar and peroneal nerves, with normal motor nerve conduction velocities (MCV). Needle examination showed fibrillations, positive sharp waves and widespread polyphasic motor unit potentials in the genioglossus and tested muscles of the upper and lower limbs. Serum paraneoplastic panel examination tested positive for antibodies against Sry-like high mobility group box 1 (SOX1, titer 1:100) and glutamic acid decarboxylase 65 (GAD65, titer 1:32). However, whole-body PET-CT showed no tumors. He received intravenous immunoglobulin (IVIG, 0.4 g/kg/d) for 5 days and riluzole 50 mg twice daily. However, his symptoms did not improve and continued to deteriorate. He died of respiratory failure one year later.

Case 2: A 47-year-old male presented with a 13-month history of muscle weakness in the left lower limb. He had received partial resection of the right kidney due to a right renal clear cell carcinoma 11 months earlier, followed by diffuse muscle twitches and aggravated weakness of the left lower limb. Neurological examination revealed atrophy of the bilateral thenar muscles, diffuse muscles weakness, pathologically brisk limb reflexes and bilaterally positive Babinski signs. EMG showed extensively acute and chronic denervation (fibrillations and positive sharp waves, increased amplitude and duration of motor unit action potentials, and reduced recruitment pattern) in the bulbar, cervical and lumbosacral regions. Brain and spine MRI showed no obvious abnormality. The paraneoplastic antibodies (anti-Hu, -Yo, -Ri, -SOX1, -Titin, -Recoverin, -Ma2/Ta, -Ma1, -CV2, -amphiphysin, -GAD65, -Tr, -Zic4 and -PKCγ antibodies) in both serum and cerebrospinal fluid (CSF) were negative, and CSF analysis showed normal pressure (120 mmH_2_O), 1.00 × 10^6^/L nucleated cells and 536.6 mg/L protein. Subsequently, the patient was treated with 0.4 g/kg/d IVIG for 5 days, a tapering dose of oral prednisolone (initial dose: 60 mg per day) and 50 mg riluzole twice daily. His muscle strength in the upper limbs improved slightly, and his clinical symptoms were stable during one-year follow-up.

Case 3: A 50-year-old male presented with 2 years of progressive lower extremity weakness. He had been diagnosed with chronic myelogenous leukemia 18 months earlier and treated with imatinib 600 mg daily. One year earlier, he had noticed atrophy of the quadriceps femoris. On neurological examination, he had diffuse muscle weakness, amyotrophy and fasciculations in lower limbs, normal tendon reflex and bilaterally positive Babinski sign. EMG showed low CMAPs in all tested nerves of the upper and lower limbs, with normal MCV. Additionally, needle examination revealed spontaneous activity (fibrillations and positive sharp waves) and widespread chronic neurogenic suffering in multiple muscles of four regions (bulbar, cervical, thoracic and lumbar regions). Brain and spine MRI were unremarkable. Routine blood analysis suggested anemia (hemoglobin 98 g/L) and normal counts of red cells, white cells, lymphocytes and blood platelets. CSF test revealed normal open pressure (150 mmH_2_O), 19.00 × 10^6^/L nucleated cells and 1184 mg/L protein. The tests for paraneoplastic antibodies (anti-Hu, -Yo, -Ri, -SOX1, -Titin, -Recoverin, -Ma2/Ta, -Ma1, -CV2, -amphiphysin, -GAD65, -Tr, -Zic4 and -PKCγ antibodies) in serum and CSF were negative. He received 80 mg/day methylprednisolone for 10 days, after which the distal muscle strength of the lower limbs improved slightly. At one-year follow-up, his clinical symptoms were not exacerbated, and the repeated CSF test indicated normal pressure (160 mmH_2_O), 10.00 × 10^6^/L nucleated cells and 496.23 mg/L protein.

## 3. Literature Review

To summarize the clinical characteristics of paraneoplastic ALS, we performed electronic searches on the PubMed data set for all papers published between 1980 and 2022 using the search terms: ((“amyotrophic lateral sclerosis” OR “ALS” OR “motor neuron” OR “motor neuron disease “ (MND) OR “neuromuscular disease”) AND (“paraneoplastic neuropathy” OR “paraneoplastic syndrome” OR “paraneoplastic neurological syndrome”)) AND ((“1980/01/01” [Date—Publication]: “2022/06/10” [Date—Publication])). The studies were screened and assessed for eligibility.

Among 72 studies obtained after the database search, 5 studies described 12 unique cases supporting the diagnosis of definite paraneoplastic ALS/MND with well-characterized onconeural antibodies present or with neurological improvement after cancer treatment (Cases 1–12 in Table 1) [7,8,9,10,11]. Three cases without onconeural antibodies but neurological symptoms remained stable after tumor resection supported the diagnosis of possible paraneoplastic ALS/MND (Cases 13–15 in Table 1). The other 15 cases, without onconeural antibodies and with cancer present within 2 years of ALS diagnosis but with progressive deterioration after tumor resection (Cases 16–30 in Table 1), are difficult to distinguish between possible paraneoplastic ALS and co-occurrence ALS and tumor [8,12,13,14,15,16,17,18,19,20,21]. Table 1 summarizes the main clinical characteristics of these 30 patients. The median age at neurological symptom onset was 61 years (range 33–84), and 17 (57%) patients were male, while 13 (43%) were female. Bulbar involvement was observed in 16 (53%) cases. A total of 28 (93%) patients had tumors, including lung (*n* = 10), breast (*n* = 4), prostate (*n* = 2), laryngeal (*n* = 2), ovarian (*n* = 1), matrix (*n* = 1), rectal (*n* = 1), stomach (*n* = 1), melanoma (*n* = 1) and lymphoma (*n* = 1). Another 2 patients with onconeural antibodies but no tumor showed stable neurological symptoms and were tumor-free during the 12- to 15-month follow-ups (Cases 1 and 10). For the 12 cases of definite paraneoplastic ALS (Cases 1–12), the median age at neurological symptom onset was 60.5 years (range 57–80), and 7 (58%) patients were male, while 5 (42%) were female. For the other 18 cases, the median age at neurological symptom onset was 61 years (range 33–84), and 10 (56%) patients were male, while 8 (44%) were female. There was no significant difference in the age of disease onset or gender distribution between the definite paraneoplastic ALS group (Cases 1–12) and the not definite paraneoplastic ALS group (Cases 13–30). Bulbar involvement was observed in 2 (17%) cases in cases 1–12 and 14 (78%) cases in cases 13–30. For patients with onconeural antibodies, the most represented onconeural antibody was anti-Hu (*n* = 9), followed by anti-Yo (*n* = 2), anti-CV2 antibodies (*n* = 2), anti-Ri (*n* = 1) and anti-CRMP5 (*n* = 1). In the definite paraneoplastic ALS cases, cancer was diagnosed after the onset of neurological symptoms in 6 (50%) cases, and 3 (25%) patients improved; 5 (42%) patients stabilized soon after the start of immunotherapy and cancer treatment. During the last follow-up (4–26 months), 1 out of 12 patients in the definite paraneoplastic ALS group (Cases 1–15) and 12 out of 18 patients in the not definite paraneoplastic ALS group died due to uncontrolled cancer or progressively deteriorated neurological symptoms. 

## 4. Discussion

Paraneoplastic ALS is a rare and non-classical form of PNS that can present as isolated or combined UMN/LMN syndrome and is challenging to distinguish from idiopathic ALS [7]. This article provides three cases, helping to broaden the understanding and enrich the existing case pool of this rare disorder.

We present the first case of paraneoplastic ALS associated with anti-SOX1 and anti-GAD65 antibodies. It has been indicated in previous reports that antibody against SOX1 is a paraneoplastic high-risk factor antibody that showed more than 70% frequency associated with cancer, and the most commonly associated cancer type with SOX1 is small cell lung cancer [6,22]. Lambert–Eaton myasthenic syndrome is the most frequent PNS found in patients positive for anti-SOX1-antibody [22,23]. Other neurological syndromes associated with anti-SOX1 antibodies have also been reported, including chronic axonal polyneuropathy, paraneoplastic limbic encephalitis and paraneoplastic cerebellar degeneration [22]; however, MND or ALS is being reported for the first time. Anti-GAD65 antibody is primarily associated with stiff-person syndrome, cerebellar ataxia, epilepsy and paraneoplastic neurological syndrome [24]. It is also linked with tumors, such as small cell lung cancer, breast cancer and thymoma [24]. Although no tumor was found in case 1 during the one-year follow-up, two positive paraneoplastic antibodies indicated the possible diagnosis of paraneoplastic ALS [6]. The patient in case 1 did not had any improvement after IVIG treatment and progressively deteriorated. The effect of the paraneoplastic antibodies in ALS pathogenesis still needs further investigation.

No paraneoplastic antibodies were found in the latter two cases, but tumors were present. Generally, tumor occurrence had a strong temporal correlation with the incidence of ALS symptoms. According to the diagnostic criteria of possible PNS (a non-classical syndrome, no onconeural antibodies and cancer present within two years of diagnosis) [6], we considered these cases possible paraneoplastic ALS. Similar to previous literature reports, these patients benefited from tumor treatment and immunotherapy, and their symptoms progressed slowly.

Literature and the current findings indicate that the onset age of paraneoplastic ALS (definite and possible) ranges from 45 to 80 years, with an average of 61 years. There is no significant difference in gender distribution in paraneoplastic ALS. The most common tumor types are lung, breast, prostate and laryngeal. Notably, bulbar involvement was not frequent in patients with paraneoplastic ALS. For patients with onconeural antibodies, the most represented antibody was anti-Hu, followed by anti-Yo, anti-CV2, anti-Ri and anti-CRMP5. The common clinical manifestations include limb weakness, general fatigue, weight loss, bulbar dysfunction and muscle cramp. A few cases also experienced subacute depression, erectile dysfunction, anorexia and diaphragmatic paralysis. Sixty percent of the paraneoplastic ALS patients could achieve improvement or remained stable soon after cancer treatment and immunotherapy. For the progressively deteriorated patients after cancer treatment, it is hard to distinguish whether the ALS is tumor origin or is just coincidental with co-occurrent tumor. However, it is important to correctly identify ALS patients with paraneoplastic origin, as most of them can improve or at least stabilize following immunotherapy and tumor treatment. 

This study reports the first paraneoplastic ALS case with anti-SOX1 and anti-GAD65 antibodies. The case series provides a better understanding of this unique type of ALS. We recommend increased attention to screening tumors and paraneoplastic antibodies in suspected ALS patients. When tumors or paraneoplastic antibodies are detected, then immunotherapy should be considered. Additionally, future studies should prioritize clinical research that explores the correlation between ALS and cancer. 

## Figures and Tables

**Table 1 brainsci-12-01053-t001:** Review of the cases of definite paraneoplastic MND and possible paraneoplastic ALS.

No. Study	No. Case	Age (year)/Gender	Diagnosis from Symptom Onset Mean Time (months)	Initial Neurological Symptoms	ALS Classification at Diagnosis	Bulbar Involvement	Electrophysiological Features at Diagnosis	Paraneoplastic Antibodies	Type of Neoplasm and Treatment	Diagnosis from Cancer to Onset of ALS Time	Treatment Response	Prognosis
1. Assaf Tolkovsky. 2021 [7]	Case 1	72/M	0.5	subacute depression, erectile dysfunction, anorexia and weight loss	Definite ALS	Yes	upper-limb fasciculations and fibrillations, decreased recruitment and increased duration and amplitude of motor unit action potentials	anti-Hu, anti-CV2/CRMP5	None	N.A.	little effect (immunosuppressive/immunomodulatory treatment)	Stable for 15 months
2. John Goodfellow et al., 2019 [8]	Case 2	57/F	1	weight loss and weakness of her right arm	Definite ALS	No	active denervation in all 5 upper limb muscles tested bilaterally	anti-Hu, anti-CV2	small cell lung cancer, ChT	At the same time	Improvement	regaining weight and strength
3. Shamsuddin Khwaja et al., 2020 [9]	Case 3	67/F	4	progressive weakness and weight loss	Definite ALS	No	positive sharp waves fibrillation potentials	anti-Yo	ovarian carcinoma	At the same time	Progressively deteriorated	Progressive course
4. Capucine Diard-Detoeuf et al., 2014 [10]	Case 4	80/F	1	bilateral upper extremity weakness	UMN dominant	No	denervation in the upper limbs and motor unit potentials in several leg muscles.	anti-Ri	breast cancer, HT, surgery, RT, IVIG, corticosteroids	4 years before the onset of ALS	a moderate clinical improvement	continued improvement
5. Nicolas Mélé et al., 2018 [11]	Case 5	60/M	/	weakness of the four limbs and tetrapyramidal signs	Probable ALS and cerebellar syndrome	No	Myotome S1 of both LL	anti-Yo	Prostatic adenocarcinoma, surgery, HT, RT, IVIG	1.5 year after the onset of ALS	no exacerbations of the neurological symptoms	Stable for 2 years
	Case 6	64/F	/	weakness of the upper limbs	LMN dominant	No	Widespread denervation of both UL	anti-Hu	Metastatic breast carcinoma, ChT, RT, IVIG, Cyclo	1 year before the onset of ALS	progressively deteriorated	Progressive course
	Case 7	64/M	/	weakness of the upper limbs, diaphragmatic paralysis and tetrapyramidal signs	Probable ALS	No	Widespread denervation of both UL	anti-Hu	Squamous cell lung carcinoma, surgery, ChT, IVIG	Concomitant to relapse	progressively deteriorated	Oncological worsening leading to death after 4 months
	Case 8	60/M	/	weakness of the lower limbs	LMN dominant	No	Denervation in the left LL (L5) and the right LL (S1)	anti-Hu	Small cell lung cancer, ChT, IVIG	3 months after the onset of ALS	no exacerbations of the neurological symptoms	Stable for 2 years
	Case 9	59/M	/	weakness of the upper limbs	LMN dominant	No	Widespread denervation in the left UL	anti-Hu	Small cell lung cancer, ChT, RT, IVIG, Corticosteroids	2 months after the onset of ALS	Significant improvement	Significant improvement
	Case 10	59/M	/	weakness of the four limbs (predominant in the upper limbs)	LMN dominant	No	Predominant denervation in the left UL	anti-Hu	None, IVIG	N.A.	no exacerbations of the neurological symptoms	Stable for 1 year
	Case 11	59/M	/	weakness of the upper limbs and tetrapyramidal signs	Probable ALS	No	Predominant denervation in the right UL	anti-Hu	Small cell lung cancer, ChT, RT, IVIG, Cyclo	6 months after the onset of ALS	no exacerbations of the neurological symptoms	Stable for 1 year
	Case 12	61/F	/	weakness of the upper limbs, diaphragmatic paralysis, and bulbar dysfunction	Probable ALS	Yes	Widespread denervation of both UL	anti-Hu	Squamous cell lung carcinoma, surgery, IVIG, Corticosteroids	4 months befor the onset of ALS	no exacerbations of the neurological symptoms	Stable for 1 year
6. Yasunao Kogashiwa et al., 2011 [12]	Case 13	58/M	<17	general fatigue and a sudden weakness in his lower limbs	probable ALS	Yes	normal nerve conduction and no abnormal findings	N.A.	hypopharyngeal squamous cell carcinoma, ChT, RT	22 months before the diagnosis of ALS	no exacerbations of the neurological symptoms	Stable for 13 months with tumor-free since his last radiation treatment
7. Onur Akan et al., 2021 [13]	Case 14	45/M	12	Painful cramps at legs, right leg weakness	LMN dominant, flail leg syndrome	No	Widespread denervation and chronic neurogenic changes in lumber region	N.A.	Non-small cell lung carcinoma with lymph node metastasis operated, RT, ChT, IVIG	3 years after the onset of ALS	no exacerbations of the neurological symptoms	Stable for 1 year
8. Yuko kijima et al., 2005 [14]	Case 15	52/F	72	general fatigue and a sudden weakness in her upper limbs	Definite ALS	No	high amplitude motor unit potential, fibrillation potential	Negative	breast cancer, surgery	6 years after the diagnosis of ALS	no exacerbations of the neurological symptoms	Stable for 1 year
9. Agnes Mondok et al., 2010 [15]	Case 16	66/F	24	moderate dysarthria	Definite ALS	Yes	N.A.	N.A.	pituitary adenoma, ductal invasive carcinoma, HT, surgery	18 months before the onset of ALS	progressively deteriorated	died 21months after ALS diagnosis (a sudden respiratory arrest)
10. Yasushi SATO et al., 2007 [16]	Case 17	60/M	5	Grip strength weakness	Definite ALS	Yes	Broad polyphasic high amplitude potential	N.A.	rectal cancer (Stage IV), palliative surgery, ChT	8 months before the definition of ALS	significant tumor reduction, but ALS symptoms progressively deteriorated	died 3 months after ALS diagnosis (aspiration pneumonia)
11. John Goodfellow et al., 2019 [8]	Case 18	61/F	9	walking difficulties with weakness and stiffness in her legs.	Definite ALS	No	fibrillations and positive sharp waves in multiple lower limb muscles bilaterally	Negative	ductal carcinoma of breast, surgery, RT, HT	At the same time	progressively deteriorated	died 26 months after symptom onset
12. Takahiro Shiba et al., 2021 [17]	Case 19	69/F	4	weakness in the toes	Definite ALS	Yes	denervation in three areas of the spinal cord	N.A.	mesenteric lymphoma	2 months before the onset of ALS	progressively deteriorated	Died 2 months after ALS diagnosis (systemic inflammatory response syndrome)
13. Masoud Mehrpour et al., 2013 [18]	Case 20	79/F	2	dysphagia	Definite ALS	Yes	low CMAPs active denervation	N.A.	neuroendocrine tumor of stomach	At the same time	N.A.	N.A.
14. Muhammad Jaffer et al., 1998 [19]	Case 21	63/M	3	bradykinesia, shuffling gait, muscle cramping, and foot drop	Definite ALS	Yes	widespread neuropathic process with diffuse denervation and re-innervation	N.A.	metastatic melanoma	N.A.	dramatically worsened (immunotherapy)	died from acute hypoxic respiratory failure rapidly
15. Onur Akan et al., 2021 [13]	Case 22	78/M	18	Painful cramps at legs, bilateral leg weakness	LMN dominant, PMA	Yes, developed in the course of the disease	Widespread denervation and chronic neurogenic changes in cervical, thoracic and lumber regions and sensory PNP	N.A.	Prostate adenocarcinoma, TURP	At the same time	progressively deteriorated	Died 9 months after ALS diagnosis
	Case 23	84/F	5	Hypophonia, dysarthria, painful cramps at right arm	Definite ALS	Yes, at the onset of the disease	Widespread denervation and chronic neurogenic changes in cervical and lumber regions	N.A.	Endometrial serous adenocarcinoma, RT	20 months before the onset of ALS	progressively deteriorated	Died 9 months after ALS diagnosis
	Case 24	61/M	12	Painful cramps, left arm weakness	Definite ALS	Yes, developed in the course of the disease	Widespread denervation and chronic neurogenic changes in cervical, thoracic and lumber regions	N.A.	Lung adenocarcinoma with lymph node metastasis, RT, ChT	1 year after the onset of ALS	progressively deteriorated	Died 8 months after ALS diagnosis
	Case 25	66/M	48	Right leg weakness with spasticity	Definite ALS	Yes, developed in the course of the disease	Widespread denervation and chronic neurogenic changes in cervical, thoracic and lumber regions	N.A.	Laryngeal squamous cell carcinoma, surgery RT	2 years before the onset of ALS	progressively deteriorated	Died 2 years after ALS diagnosis
	Case 26	33/F	12	Painful cramps at right arm, right arm weakness	Probable ALS	Yes, developed in the course of the disease	Widespread denervation and chronic neurogenic changes in cervical region	N.A.	MGUS, plasmapheresis	1 year after the onset of ALS	progressively deteriorated	Progressive course
16. Xiaochuang Yuan et al., 2017 [20]	Case 27	42/M	6	Lower extremity weakness	Definite ALS	No	Widespread denervation in cervical, thoracic and lumber regions with fasciculation potential and positive sharp wave	N.A.	pituitary adenoma, HT, surgery	6 months after the onset of ALS	Deterioration of muscle weakness	Progressive course
17. Guangyou Sun et al., 1995 [21]	Case 28	56/M	6	Muscle weakness and atrophy in both upper limbs	Definite ALS	Yes	Widespread denervation	N.A.	Central lung cancer (SCC)	6 months after the onset of ALS	progressively deteriorated	Died 2 months after ALS diagnosis (dyspnea, pulmonary infection)
	Case 29	64/F	5	dysarthria, dysphagia	Definite ALS	Yes	Neuronal damage	N.A.	Mediastinal mass	5 months after the onset of ALS	progressively deteriorated	Died 6 months after ALS diagnosis (dyspnea, pulmonary infection)
	Case 30	61/M	8	Muscle weakness and atrophy of the right upper limb	Definite ALS	Yes	Neuronal damage	N.A.	Central lung cancer	8 months after the onset of ALS	progressively deteriorated	Died 3 months after ALS diagnosis (dyspnea, dysphagia, pulmonary infection)

ALS: Amyotrophic lateral sclerosis, ChT: chemotherapy, Cyclo: cyclophosphamide, F: female, IVIG: intravenous immunoglobulin, HT: Hormonal therapy, LMN: lower motor neuron, M: male, MGUS: Monoclonal gammopathy of unknown significance, RT: radiotherapy, TURP: transurethral resection of the prostate.

## Data Availability

Not applicable.

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
