# Peer review of "Paraneoplastic Amyotrophic Lateral Sclerosis: Case Series and Literature Review"

_brainsci, 2022, doi:10.3390/brainsci12081053_

Round 1

Reviewer 1 Report

In this study, the authors presented three cases of possible paraneoplastic ALS. They also reviewed the literature on paraneoplastic ALS and summarized the clinical features, treatment effects, and prognosis of these patients. The study is interesting and well-written I only have some minor concerns.

1.      The authors diagnosed the three cases as possible paraneoplastic ALS. I would suggest the authors present the diagnostic criteria of paraneoplastic ALS in the part of the literature review.

2.      The authors reviewed the literature and identified 30 cases, 15 cases matched the criteria for paraneoplastic ALS, and 15 cases were ALS patients with co-occurrent tumors that showed no onconeural antibodies or progressively deteriorated after tumor resection. I would like to query why they diagnosed case 2 and case 3 in their case series as possible paraneoplastic ALS rather than ALS patients with co-occurrent tumors.

3.      In case2, EMG showed extensive damage to the anterior horns in the bulbar, cervical, and lumbosacral regions. Actually, it is difficult to differentiate lesions of anterior horns from lesions of anterior roots. I would like to query how did the authors determine the lesions of anterior horns.

4.      In case 3, EMG showed low CMAPs in all tested nerves of the upper and lower limbs, with normal MCV. Additionally, needle examination revealed spontaneous activity (fibrillations and positive sharp waves) in all tested muscles. I would like to query if they also examined the muscles of regions of bulbar and thoracic. 

Reviewer 2 Report

The manuscript summarizes the current literature about paraneoplastic ALS, discussing the clinical features, treatment effects, and prognosis of the existing cases. Moreover, the authors describe three new cases of possible paraneoplastic ALS.

The manuscript is well-written and concise. However, in my opinion, some additional references could be added, and the order of the paragraphs changed to make the text more comprehensive. Below is some advice.

Abstract

Line 16. “And we reported two ALS..” I suggest not starting a sentence with "and" following a point.

1. Introduction

Lines 25-27: Please add pertinent citation(s)

Lines 27-30: Please add pertinent citation(s)

2. Cases series

Line 57: “The paraneoplastic antibodies…” Which antibodies? Please clarify, or cite the tested antibodies

Line 77: “paraneoplastic and ganglioside antibodies…” Same as the comment for line 57

3. Literature review

I appreciate the systematic classification done to catalog the already known paraneoplastic ALS cases. Moreover, also the table helps in summarizing them.

I suggest moving the literature review firstly and then reporting the three new cases. In this manner, the readers can know the clinical characteristics of the known existing patients and then follow those of the new patients, getting the similarities and, most of all, the differences.

Lines 102-105: Since you specify the age of neurological symptoms onset, and the number (and percentage) of male and female patients for the "probable paraneoplastic ALS group" (cases 1-15), it could be beneficial to make the same for the "probable co-occurent ALS and tumor group" (cases 16-30) before this sentence.

In lines 87-113: sometimes the numbers of cases are indicated as numbers (e.g. “were well characterized in 12 cases”, or “17 (57%) patients were male, while 13 (43%) were female”), other time as words (e.g. “Nine (60%) patients were male while six (40%) were female”, or “in six (40%) cases, and two (13%) patients improved, nine (60%) patients stabilized soon after”. I suggest standardizing the paragraph (indicating the numbers of cases as numbers or as words).

Discussion

Lines 124-126: Please add pertinent citation(s)

Lines 126-127: Please add pertinent citation(s)

Lines 130-131: Please add pertinent citation(s)

Lines 135-137: Please add pertinent citation(s)

References

I can’t find reference 14. Please revise/correct this citation.

Reviewer 3 Report

This is an excellent review of 30 cases of likely paraneoplastic ALS, an unusual presentation of a relatively rare (but disabling and fatal for many) neurodegenerative disease . The value of this review is that it should encourage all ALS patients upon diagnosis to have onconeural antibodies assayed, to identify the few that might benefit from treatment of their malignancy. The authors have done an excellent job with a challenging literature search and provide a conservative discussion of prarneoplastic ALS.

Author Response

Dear respected reviewer,
Greetings.
Thank you very much for the precious time and effort you took on reviewing our manuscript. We appreciate your helpful comments and are pleased to have our manuscript be reviewed by you. We are hoping that you may be satisfied with our changes in this updated version. 
Thank you once again for comments on our paper.